# Effect of Postoperative Coffee Consumption on Postoperative Ileus after Abdominal Surgery: An Updated Systematic Review and Meta-Analysis

**DOI:** 10.3390/nu13124394

**Published:** 2021-12-08

**Authors:** Jun Watanabe, Atsushi Miki, Masaru Koizumi, Kazuhiko Kotani, Naohiro Sata

**Affiliations:** 1Department of Surgery, Division of Gastroenterological, General and Transplant Surgery, Jichi Medical University, Shimotsuke-City 329-0498, Japan; amiki@jichi.ac.jp (A.M.); mkoizumi@jichi.ac.jp (M.K.); sata2018@jichi.ac.jp (N.S.); 2Division of Community and Family Medicine, Jichi Medical University, Shimotsuke-City 329-0498, Japan; kazukotani@jichi.ac.jp

**Keywords:** abdominal surgery, caffeine, coffee, ileus, length of stay, meta-analysis, systematic review

## Abstract

Background: Previous systematic reviews have not clarified the effect of postoperative coffee consumption on the incidence of postoperative ileus (POI) and the length of hospital stay (LOS). We aimed to assess its effect on these postoperative outcomes. Methods: Studies evaluating postoperative coffee consumption were searched using electronic databases until September 2021 to perform random-effect meta-analysis. The quality of evidence was assessed using the Cochrane risk-of-bias tool. Caffeinated and decaffeinated coffee were also compared. Results: Thirteen trials (1246 patients) and nine ongoing trials were included. Of the 13 trials, 6 were on colorectal surgery, 5 on caesarean section, and 2 on gynecological surgery. Coffee reduced the time to first defecation (mean difference (MD) −10.1 min; 95% confidence interval (CI) = −14.5 to −5.6), POI (risk ratio 0.42; 95% CI = 0.26 to 0.69); and LOS (MD −1.5; 95% CI = −2.7 to −0.3). This trend was similar in colorectal and gynecological surgeries. Coffee had no adverse effects. There was no difference in POI or LOS between caffeinated and decaffeinated coffee (*p* > 0.05). The certainty of evidence was low to moderate. Conclusion: This review showed that postoperative coffee consumption, regardless of caffeine content, likely reduces POI and LOS after colorectal and gynecological surgery.

## 1. Introduction

Postoperative ileus (POI), defined as the transient cessation of coordinated bowel motility, is a common cause of delayed return to normal bowel function after abdominal surgery (e.g., colorectal and gynecologic surgery), occurring in 10–15% of cases [1,2]. Delayed defecation associated with POI causes vomiting, bloating, and intolerance to food, and POI often leads to invasive interventions, such as nasogastric tube insertion [3]. POI increases postoperative length of hospital stay (LOS) and treatment-related costs [4,5]. POI and LOS are important postoperative outcomes because prolonged LOS and increased risk of morbidity due to POI have been shown to reduce patients’ quality of life and increase hospital expenditures [4,5,6].

Coffee is the most widely consumed pharmacological substance worldwide [7]. Caffeine exerts anti-inflammatory effects on the gastrointestinal and cardiovascular systems, mediated by its antagonistic effects on A2A receptors on immune cells, such as T and B cells and macrophages [8,9]. Since the implementation of enhanced recovery protocols (ERPs), multimodal strategies have been used to improve the postoperative return of gastrointestinal function [10,11]. Recommendations regarding the use of postoperative coffee vary in various international ERPs [10,11]. Previous systematic reviews did not demonstrate that LOS and POI were statistically significantly reduced, because of the small number of trials [12,13,14,15]. In addition, it is unclear whether coffee or decaffeinated coffee is effective in treating POI [12].

Coffee, a popular and easily available beverage worldwide, could also be clinically significant if shown to prevent POI incidence in addition to shortening LOS. In terms of ERPs, colorectal and gynecological surgeries are treated similarly because of the manipulation of the bowel [10,11]. Therefore, the present updated systematic review and meta-analysis aimed to assess the effect of postoperative coffee consumption on POI after abdominal surgery, including colorectal surgery, cesarean section, and gynecological surgery.

## 2. Material and Methods

### 2.1. Protocol

We followed the Preferred Reporting Items for Systematic Review and Meta-Analysis 2020 (PRISMA 2020) (Appendix A) [16]. This protocol was registered on protocols.io (https://doi.org/10.17504/protocols.io.bymmpu46).

### 2.2. Inclusion Criteria

Randomized controlled trials (RCTs) that assessed the effect of postoperative coffee consumption after abdominal surgery were included. No language, country, observation period, or publication year restrictions were applied. Review articles, case series, and case reports were excluded. The intervention of interest was postoperative 100–150 mL coffee consumption, three times per day, for 10–20 min. The control group consumed water, tea, or a placebo. The primary outcomes were time to first defecation (hours), LOS (days), and POI. The secondary outcomes were the time to first flatus (hours), the time to first bowel movement (hours), the time to tolerance of solid food (hours), and adverse events.

### 2.3. Search Method

The following electronic databases and trial registries were searched: MEDLINE (PubMed), Cochrane Central Register of Controlled Trials (Cochrane Library), EMBASE (Dialog) (Appendix B), the World Health Organization International Clinical Trials Platform Search Portal (ICTRP), and ClinicalTrials.gov (Appendix C). The reference lists were checked for studies, including international guidelines [10,11], as well as reference lists of eligible studies and articles citing eligible studies. The authors of the original studies were asked for unpublished or additional data if necessary.

### 2.4. Data Collection and Analysis

Two independent reviewers (J.W. and A.M.) performed screening, data extraction, and assessment of the risk of bias using the Risk of Bias 2 tool [17] and assessed the quality of evidence based on the Grading of Recommendations Assessment, Development and Evaluation (GRADE) approach [18]. Disagreements between the two reviewers were discussed, and if necessary, a third reviewer (K.K.) was consulted.

The relative risk ratios (RRs) and the 95% confidence intervals (CIs) were calculated for the binary variables, POI, and adverse events. The mean differences (MDs) and 95% CIs were calculated for the continuous variables, LOS (days), the time to first defecation (hours), the time to first flatus (hours), the time to first bowel movement (hours), and time to tolerance of solid food (hours). Intention-to-treat analysis was performed for dichotomous data as far as possible. For continuous data, missing data were not imputed based on the recommendations of the Cochrane handbook [19]. In cases where missing data were not known after contacting the original authors, the standard deviation was calculated using the method provided in the Cochrane handbook [19] or a previously validated method [20]. A random-effects meta-analysis was performed using Review Manager software (RevMan 5.4.2).

### 2.5. Assessment of Heterogeneity and Reporting Bias

Statistical heterogeneity was evaluated by visual inspection of the forest plots and calculating the I^2^ statistic (I^2^ = 0–40%, might not be important; 30–60%, moderate heterogeneity; 50–90%, substantial heterogeneity; and 75–100%, considerable heterogeneity) [19]. When there was substantial heterogeneity (I^2^ > 50%), we assessed the reason for the heterogeneity. Cochrane’s chi^2^ test (Q-test) was performed on the I^2^ statistic, and a *p*-value less than 0.10 was defined as statistically significant. We searched the clinical trial registry system (ClinicalTrials.gov and ICTRP) to assess any reporting bias. Potential publication bias was evaluated through visual inspection of the funnel plots.

### 2.6. Additional Analysis

The following subgroup analyses were performed: surgery types (colorectal resection, cesarean section, or gynecological resection) and coffee types (caffeinated or decaffeinated coffee). The following sensitivity analysis was performed: exclusion of studies using imputed statistics.

## 3. Results

Figure 1 shows the study search process. After the removal of duplicates, 1005 records were screened, of which 31 underwent full-text review and 1 article was added after reviewing reference lists. Finally, 27 studies were included in the qualitative synthesis. The 27 studies comprised 9 ongoing trials (NCT 02510911, NCT02639728, NCT03143621, NCT03191877, NCT03712891, NCT04205058, NCT04547868, IRCT20200116046153N1, and CTRI/2021/04/033141), 5 protocols without results (NCT00130026, NCT01130675, NCT02250924, NCT03660267, and NCT03815877), and 13 clinical trials (1246 patients) [21,22,23,24,25,26,27,28,29,30,31,32,33].

Table 1 shows the characteristics of the included clinical trials. Of the 13 trials [21,22,23,24,25,26,27,28,29,30,31,32,33], 6 were on colorectal surgery, 5 on cesarean section, and 2 on gynecological surgery. The intervention was caffeinated coffee in 10 trials and decaffeinated coffee in 3 trials.

The risk of bias is shown in Table 2 and Appendix D and Appendix E. In terms of the overall risk of bias for the time to first defecation, there were concerns about the risk of bias for most studies (11/13), with two of these assessed as having a high risk of bias [25,26].

Table 3 summarizes the findings of the GRADE approach. The certainty of the evidence was low to moderate due to the high risk of bias and inconsistency.

### 3.1. Primary Outcomes

#### 3.1.1. Time to First Defecation (Hours)

Coffee reduced the time to first defecation after colorectal surgery (MD −15.37 h; 95% CI = −18.0 to −12.75; I^2^ = 0%) and gynecological surgery (MD −12.83 h; 95% CI = −22.44 to −3.23; I^2^ = 92%) but not after cesarean section (MD −4.79 h, 95% CI = −10.32 to 0.74; I^2^ = 94%) (Figure 2).

#### 3.1.2. LOS (Days)

Coffee reduced LOS after gynecological surgery (MD −1.08 d; 95% CI = −1.63 to −0.54; I^2^ = 0%) but not after colorectal surgery (MD −1.78 d; 95% CI = −4.31 to 0.75; I^2^ = 99%) and cesarean section (MD −0.30 d; 95% CI = −0.70 to 0.10; I^2^ = 93%) (Figure 3).

#### 3.1.3. POI

Coffee reduced POI incidence after cesarean section (RR 0.32; 95% CI = 0.14 to 0.72) and gynecological surgery (RR 0.25; 95% CI = 0.13 to 0.48; I^2^ = 0%) but not after colorectal surgery (RR 0.81; 95% CI = 0.40 to 1.63; I^2^ = 0%) (Figure 4).

### 3.2. Secondary Outcomes

#### 3.2.1. Time to First Flatus (Hours)

Coffee reduced the time to first flatus after abdominal surgery (MD −4.27 h; 95% CI = −8.28 to −0.26; I^2^ = 96%) (Figure A1). There was no statistically significant difference between colorectal surgery, cesarean section, or gynecological surgery in the subgroup test (*p* = 0.36).

#### 3.2.2. Time to First Bowel Sound (Hours)

Coffee reduced the time to first flatus after gynecological surgery (MD −8.87 h; 95% CI = −14.65 to −3.09; I^2^ = 86%) but not after cesarean section (MD −1.87 h; 95% CI = −4.40 to 0.66; I^2^ = 93%) (Figure A2).

#### 3.2.3. Time to Tolerance of Solid Food (Hours)

Coffee reduced the time to tolerance of solid food after colorectal surgery, cesarean section, and gynecological surgery (MD −10.11 h; 95% CI = −14.26 to −5.95; I^2^ = 95%) (Figure A3).

#### 3.2.4. Complications/Adverse Events

There have been no reports of adverse events related to postoperative coffee consumption. Coffee did not increase the risk of complications or adverse events after colorectal surgery (RR 0.85; 95% CI = 0.48 to 1.51; I^2^ = 40%) and cesarean section (RR 0.80; 95% CI = 0.23 to 2.81). Coffee decreases complications after gynecological surgery (RR 0.27; 95% CI = 0.13 to 0.53) (Figure A4).

### 3.3. Additional Analyses

In subgroup analyses of caffeinated vs. decaffeinated coffee (Figure A5, Figure A6, Figure A7, Figure A8, Figure A9, Figure A10 and Figure A11), there were statistically significant differences between caffeinated and decaffeinated coffee for the time to first defecation (*p* = 0.02) and the time to tolerance of solid food (*p* = 0.04). However, when analyzed by surgery type, there were no statistically significant differences between caffeinated and decaffeinated coffee for the time to first defecation after colorectal surgery (*p* = 0.14) or cesarean section (*p* = 0.51) (Figure A12) or for the time to tolerance of solid food after cesarean section (*p* = 0.35) (Figure A13). The results of the sensitivity analysis, excluding studies using imputed statistics, were consistent with the original results except for the time to first flatus (Figure A14, Figure A15 and Figure A16).

Regarding publication bias, the funnel plots were symmetric, suggesting a no-potential-no-publication bias (Figure A17).

## 4. Discussion

This systematic review and meta-analysis demonstrated that postoperative coffee consumption likely reduces the time to first defecation, LOS, and POI after abdominal surgery. This trend is similar to the trends after colorectal and gynecological surgeries. Additionally, there was no difference in LOS and POI between caffeinated and decaffeinated coffee intake. This updated evidence is beneficial to both patients and surgeons regarding the practical endpoints of LOS and POI.

In previous systematic reviews [12,13,14,15], coffee accelerated the postoperative recovery of gastrointestinal function but did not reduce POI and LOS. The present review in 13 RCTs with 1246 patients extends the findings of previous reviews, showing a novel benefit of coffee for POI and LOS, in addition to standard ERPs. Preventing POI and shortening LOS can potentially affect the quality of life of patients and reduce their social costs by approximately 40–50% per patient [4,5,6]. In addition, preventing POI and shortening LOS has the potential to reduce hospital expenditures by US$750 million per year [4,5]. On average, the incidence of POI was 60% lower in the coffee group (POI: 6.9%) than in the non-coffee control group (16.5%). With postoperative coffee consumption, LOS was reduced by 1.5 days. Given that other consensus data show that ERPs reduce morbidity (RR 0.78) and LOS (−3.1 days) and opioid antagonists, which are frequently used to improve the postoperative course, reduce POI (32%) and LOS (−0.3 days) [34,35], the improved POI and LOS following coffee intake appear to be meaningful in the clinical setting.

The mechanism underlying the effect of coffee on POI is not fully understood. The factors may be caffeine and other substances in coffee, mainly phenolic antioxidants of chlorogenic acid [36]. Caffeine acts positively on inflammation, activating ryanodine-sensitive Ca^2+^ channels by releasing Ca^2+^ from the sarcoplasmic reticulum and inhibiting cyclic guanosine monophosphate degradation, thereby promoting nitric oxide synthesis in the endothelium and enhancing caffeine-induced endothelium-dependent vasodilation [37,38,39]. Caffeine promotes postoperative recovery of gastrointestinal function through vasodilation [32,40]. Chlorogenic acid has beneficial effects on inflammation and pain [41]. Chlorogenic acid has an anti-inflammatory effect by potently inhibiting the production of tumor necrosis factor-α and interleukin-6 by peripheral blood mononuclear cells [42,43]. In addition, chlorogenic acid inhibits edema formation leading to pain and improves pain following inflammatory responses [42]. These effects may prevent POI and/or lead to shorter LOS.

There were no differences in the recovery of postoperative gastrointestinal function between caffeinated and decaffeinated coffee. These results suggest that caffeine and non-caffeine substances may have a positive effect on POI. In previous studies, both caffeinated and decaffeinated coffee similarly reduced the risk of various cancers and death from all causes [44,45]. The results of our study were in accordance with those of previous studies. However, caution should be exercised when interpreting the results due to the small number of studies involving decaffeinated coffee.

In the present review, there were no reports of adverse events related to coffee, although the caffeine group had a higher postoperative systolic blood pressure (mean 120 mmHg) than that of the control group (mean 100 mmHg) [32,46]. The amount of coffee used in this study was a common amount, and considering the safety of coffee, which is widely used, it is not a phenomenon that should be of great concern [47]. Whether hypertensive patients need to refrain from coffee consumption after surgery requires further study.

Our study showed that the certainty of the evidence was low to moderate because of the high risk of bias and inconsistency based on the GRADE approach. The overall risk of bias was high because the concealment of the allocation sequence was unclear, and the outcomes of interest, POI and LOS, were not included in the protocol. Further studies are needed to clarify allocation concealment and clarify outcomes, such as POI and LOS, in protocols. Additionally, the definitions of POI and LOS were unclear and may be affected by blinding and socioeconomic confounds. In the present review, many studies reported that POI was the indication for reinsertion of the nasogastric tube. POI and LOS should be clearly defined and recorded by blinded outcome assessors. When interpreting our results, heterogeneity in variables such as age, comorbidities, and surgical invasiveness in each population undergoing the procedure should be considered. In the case of cesarean section, the impact of coffee on LOS after cesarean section may be small because the hospital stay is short to begin with [48,49]. In the case of colorectal surgery, coffee had a relatively weak effect on POI, which may be due to other factors related to POI, such as postoperative exercise and nutrition [35,50].

This review has additional limitations. First, the dose–response relationship between coffee consumption and outcomes was not evaluated. In the studies included in this review, the amount of coffee consumed was 100–150 mL, three times per day over 10–20 min. Second, the characteristics of coffee consumers, such as the relationship between regular and non-regular coffee drinkers, have not been clearly reported. Third, our results may not be generalizable to all populations because the compounds in coffee may vary by region, bean type, roast, and brewing method. Furthermore, none of the studies included data collected from children or low-income countries.

## 5. Conclusions

The findings of this updated systematic review and meta-analysis indicate that postoperative coffee consumption, with or without caffeine consumption, may reduce POI and LOS after colorectal surgery, cesarean section, and gynecological surgery. The findings suggest that patients and surgeons should preferably use postoperative coffee to reduce POI. More RCTs are needed to verify the effect of postoperative coffee consumption because the evidence for its consumption is limited by variations in surgeries.

## Figures and Tables

**Figure 1 nutrients-13-04394-f001:**
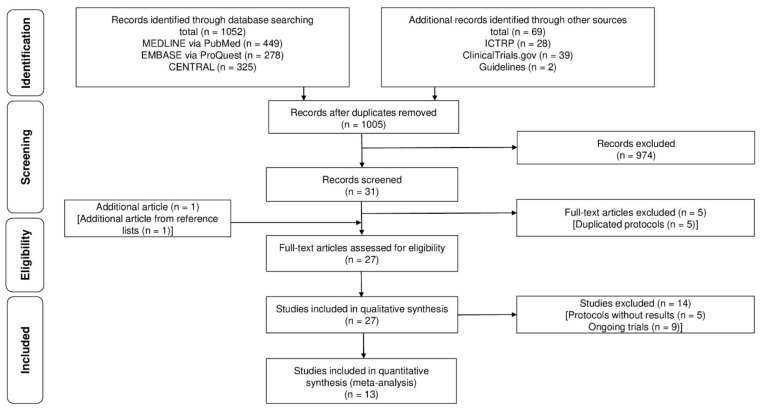
Flow of the study search process.

**Figure 2 nutrients-13-04394-f002:**
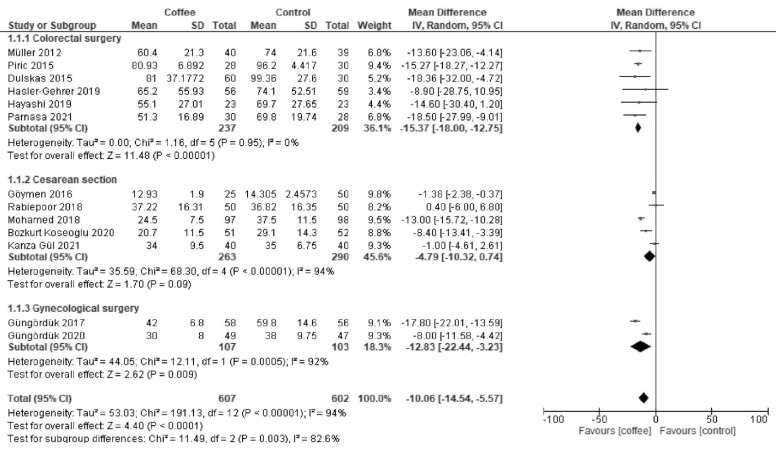
Forest plot of the time to first defecation.

**Figure 3 nutrients-13-04394-f003:**
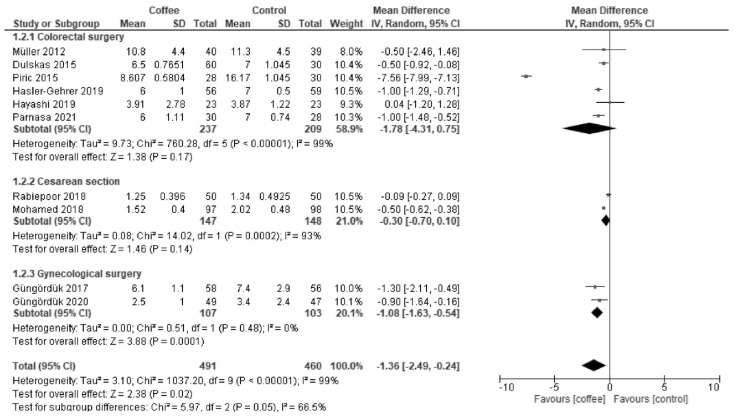
Forest plot of the length of hospital stay.

**Figure 4 nutrients-13-04394-f004:**
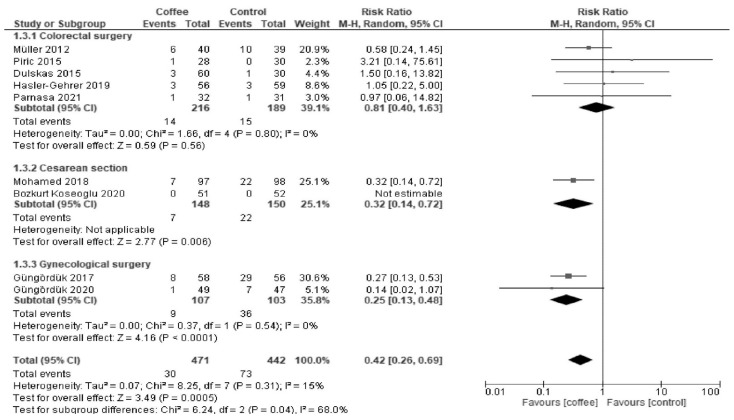
Forest plot of postoperative ileus.

**Table 1 nutrients-13-04394-t001:** The characteristics of the included studies.

Authors [ref. no.]	Year	Country	No.	Age, Year	Male, %	Surgical	Coffee	Volume, mL	Frequency	Control
Müller [21]	2012	Germany	79	61	56	CRS	Caffeine	100	TDS	Water
Dulskas [22]	2015	Lithuania	90	65	53	CRS	Caffeine, Decaf	100	TDS	Water
Piric [23]	2015	Bosnia and Herzegovina	58	63	59	CRS	Caffeine	100	TDS	Tea
Göymen [24]	2016	Turkey	75	50	0	CS	Decaf	100	TDS	Water, no intervention
Güngördük [25]	2017	Turkey	114	55	0	GS	Caffeine	100	TDS	No intervention
Mohamed [26]	2018	Egypt	210	NR	0	CS	NR	NR	NR	No intervention
Rabiepoor [27]	2018	Iran	100	28	0	CS	Caffeine	100	TDS	Water
Hasler-Gehrer [28]	2019	Switzerland	115	66	51	CRS	Caffeine	150	TDS	Tea
Hayashi [29]	2019	Japan	46	77	26	CRS	Caffeine	100	TDS	Water
Bozkurt Koseoglu [30]	2020	Turkey	113	29	0	CS	Caffeine	100	TDS	No intervention
Güngördük [31]	2020	Turkey	96	60	0	GS	Caffeine	150	TDS	Water
Kanza Gül [32]	2021	Turkey	80	28	0	CS	Decaf	NR	TDS	No intervention
Parnasa [33]	2021	Israel	70	56	50	CRS	Caffeine	50 *	TDS	Placebo

CRS, colorectal surgery; CS, caesarean section; GS, gynecological surgery; No., number; NR, not reported; TDS, three times per day. * 100 mg of caffeine citrate.

**Table 2 nutrients-13-04394-t002:** Risk of bias for the eligibility studies for the time to first defecation.

Authors [ref. no.]	Risk of Bias 2 Tool Assessment					
	Bias Arising from the Randomization Process	Bias Due to Deviations from Intended Interventions	Bias Due to Missing Outcome Data	Bias in the Measurement of the Outcome	Bias in the Selection of the Reported Results	Overall Risk of Bias
Müller [21]	Low	Some concerns	Some concerns	Some concerns	Some concerns	Some concerns
Dulskas [22]	Some concerns	Some concerns	Some concerns	Some concerns	Some concerns	Some concerns
Piric [23]	Some concerns	Some concerns	Some concerns	Some concerns	Some concerns	Some concerns
Göymen [24]	Some concerns	Low	Low	Some concerns	Some concerns	Some concerns
Güngördük [25]	Some concerns	Low	Low	Some concerns	High	High
Mohamed [26]	Some concerns	High	High	Some concerns	Some concerns	High
Rabiepoor [27]	Some concerns	Low	Low	Some concerns	Some concerns	Some concerns
Hasler-Gehrer [28]	Low	Some concerns	Some concerns	Some concerns	Low	Some concerns
Hayashi [29]	Low	Low	Low	Some concerns	Low	Some concerns
Bozkurt Koseoglu [30]	Low	Some concerns	Some concerns	Some concerns	Low	Some concerns
Güngördük [31]	Low	Some concerns	Some concerns	Some concerns	Low	Some concerns
Kanza Gül [32]	Low	Low	Low	Some concerns	Some concerns	Some concerns
Parnasa [33]	Low	Some concerns	Some concerns	Some concerns	Low	Some concerns

**Table 3 nutrients-13-04394-t003:** Summary of findings.

Effect of Postoperative Coffee Consumption after Abdominal Surgery						
Patient: Adults after Abdominal Surgery; Setting: In-Patients; Intervention: Coffee; Comparison: Control						
Outcomes	Anticipated Absolute Effects * (95% CI)		Relative Effect (95% CI)	Patient Number (Studies)	Certainty of the Evidence (GRADE)	Comments
	Risk with control	Risk with coffee				
Time to first defecation	The median time was 42 h.	MD −10 h (−14 to −5.6)	-	1209 (13 RCTs)	Moderate ^a^	Coffee reduced the time to first defecation.
Length of hospital stay	The median stay was 6 days.	MD −1.5 days (−2.7 to −0.3)	-	905 (9 RCTs)	Low ^a,b^	Coffee reduced the length of hospital stay.
Postoperative ileus	165 per 1000.	69 per 1000 (43 to 114)	RR 0.42 (0.26 to 0.69)	913 (8 RCTs)	Low ^a,b^	Coffee reduced postoperative ileus.
Time to first flatus	The median time was 30 h.	MD −4.3 h (−8.5 to −0.07)	-	1113 (12 RCT)	Low ^a,b^	Coffee reduced the time to first flatus.
Time to first bowel sound	The median time was 10 h.	MD −4.3 h (−7.1 to −1.5)	-	683 (6 RCTs)	Very low ^a,b,c^	Coffee reduced the time to first bowel sound.
Time to tolerance of solid food	The median time was 48 h.	MD −9.9 h (−14 to −5.9)	-	833 (8 RCTs)	Low ^a,b^	Coffee reduced the time to first tolerance of solid food.

CI, confidence interval; MD, mean difference; RR, risk ratio. * The risk in the intervention group (and its 95% CI) is based on the assumed risk in the comparison group and the relative effect of the intervention (and its 95% CI). GRADE Working Group grades of evidence; High certainty: We are very confident that the true effect lies close to that of the estimated effect. Moderate certainty: We are moderately confident in the estimated effect. The true effect is likely to be close to the estimated effect, but there is a possibility that it is substantially different. Low certainty: Our confidence in the estimated effect is limited: The true effect may be substantially different from the estimated effect. Very low certainty: We have very little confidence in the estimated effect. The true effect is likely to be substantially different from the estimated effect. ^a^ Downgraded because of a high risk of bias. ^b^ Downgraded because of inconsistency due to substantial heterogeneity. ^c^ Downgraded because of imprecision due to the small sample size.

## Data Availability

All data relevant to the study are included in the article.

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
