# Peer review of "Effect of Postoperative Coffee Consumption on Postoperative Ileus after Abdominal Surgery: An Updated Systematic Review and Meta-Analysis"

_nutrients, 2021, doi:10.3390/nu13124394_

Round 1

Reviewer 1 Report

With interest I reviewed this meta analysis of the studies analyzing the influence of postoperative coffee consumption on postoperative ileus after abdominal surgery.

Overall, the study is well-composed. I do not have much criticism, but would like to point out some minor comments that may help improve further this comprehensive review.

In my opinion the text needs to be better phrased in the following cases:

Abstract

Lines 12-14: “Previous systematic reviews have not clarified its beneficial effect on incidence of postoperative ileus (POI), besides the lengths of hospital stay (LOS). We aimed to assess the effect of postoperative coffee consumption on postoperative outcomes.”

The above needs to be changed to: “Previous systematic reviews have not clarified the effect of postoperative coffee consumption on the incidence of postoperative ileus (POI) and the length of hospital stay (LOS). We aimed to assess its effect on these postoperative outcomes.”

Introduction

Lines 32: “delayed return” instead of “delayed in return”

Lines 38: “have been shown to cause a reduced quality of life in patients and excess hospital expenditures [4–6]”.

The above could be better rephrased to: ““have been shown to reduce patients’ quality of life and increase hospital expenditures [4–6]”.

Lines 45-48: “Recommendations regarding the use of postoperative coffee vary in various international ERPs [9,10] and previous systematic reviews have demonstrated that more clinically useful and pragmatic endpoints, such as LOS and POI, are not statistically significantly reduced because of the small number of trials [11–14].” This sentence is too long and rather fuzzy. Please consider rephrasing.

Lines 51-52: “Coffee, a popular and easily available beverage worldwide, can be considered clinically significant if it is effective in preventing POI incidence, in addition to shortening LOS.” Again, the meaning of this sentence is rather unclear. Please consider rephrasing to something like the following: “Coffee, a popular and easily available beverage worldwide, could also be clinically significant if shown to prevent POI incidence in addition to shortening LOS.”

Author Response

We greatly appreciate your review of our manuscript and the helpful suggestions. Below are our responses to the reviewers’ comments, with a description of the changes made to the manuscript.

Response to the referees

We greatly appreciate your helpful comments and suggestions.

<Reviewer 1>

With interest I reviewed this meta analysis of the studies analyzing the influence of postoperative coffee consumption on postoperative ileus after abdominal surgery.

Overall, the study is well-composed. I do not have much criticism, but would like to point out some minor comments that may help improve further this comprehensive review.

In my opinion the text needs to be better phrased in the following cases:

Abstract

Lines 12-14: “Previous systematic reviews have not clarified its beneficial effect on incidence of postoperative ileus (POI), besides the lengths of hospital stay (LOS). We aimed to assess the effect of postoperative coffee consumption on postoperative outcomes.”

The above needs to be changed to: “Previous systematic reviews have not clarified the effect of postoperative coffee consumption on the incidence of postoperative ileus (POI) and the length of hospital stay (LOS). We aimed to assess its effect on these postoperative outcomes.”

We appreciate your suggestions. We have modified the following sentences to the revised manuscript.

“Previous systematic reviews have not clarified the effect of postoperative coffee consumption on the incidence of postoperative ileus (POI) and the lengths of hospital stay (LOS). We aimed to assess its effect on these postoperative outcomes.” (on page 1, lines 12-14, in the abstract section)

Introduction

Lines 32: “delayed return” instead of “delayed in return”

We thank the reviewer for the careful review. As the reviewer suggested, we have modified “delayed return” to “delayed return”. (on page 1, lines 31, in the introduction section)

Lines 38: “have been shown to cause a reduced quality of life in patients and excess hospital expenditures [4–6]”.

The above could be better rephrased to: ““have been shown to reduce patients’ quality of life and increase hospital expenditures [4–6]”.

We thank the reviewer for the careful review. As the reviewer suggested, we have modified the following sentences to the revised manuscript.

“have been shown to reduce patients’ quality of life and increase hospital expenditures [4–6]” (on page 2, lines 37-38, in the introduction section)

Lines 45-48: “Recommendations regarding the use of postoperative coffee vary in various international ERPs [9,10] and previous systematic reviews have demonstrated that more clinically useful and pragmatic endpoints, such as LOS and POI, are not statistically significantly reduced because of the small number of trials [11–14].” This sentence is too long and rather fuzzy. Please consider rephrasing.

We appreciate your suggestions. As the reviewer suggested, we have modified the following sentence to the revised manuscript.

“Recommendations regarding the use of postoperative coffee vary in various international ERPs [9,10]. Previous systematic reviews did not demonstrate that LOS and POI were statistically significantly reduced because of the small number of trials [11–14].” (on page 2, lines 45-47, in the introduction section)

Lines 51-52: “Coffee, a popular and easily available beverage worldwide, can be considered clinically significant if it is effective in preventing POI incidence, in addition to shortening LOS.” Again, the meaning of this sentence is rather unclear. Please consider rephrasing to something like the following: “Coffee, a popular and easily available beverage worldwide, could also be clinically significant if shown to prevent POI incidence in addition to shortening LOS.”

We appreciate your suggestions. As the reviewer suggested, we have modified the following sentence to the revised manuscript.

“Coffee, a popular and easily available beverage worldwide, could also be clinically significant if shown to prevent POI incidence in addition to shortening LOS.” (on page 2, lines 49-50, in the introduction section)

Reviewer 2 Report

Manuscript:  Effect of Postoperative Coffee Consumption on Postoperative Ileus After Abdominal Surgery: An Updated Systematic Review and Meta-Analysis

Manuscript # nutrients-1444922

General Comments

This systematic review and meta-analysis examined the effect of coffee (caffeinated or decaffeinated) consumption on postoperative ileus and length of hospital stay after various types of abdominal surgeries, including colorectal resection, cesarean section, and gynecological resection. Five of the 13 included clinical trials were conducted in Turkey, with additional trials from Bosnia and Herzegovina, Egypt, Germany, Iran, Israel, Japan, Lithuania, and Switzerland. Among the primary outcomes, coffee consumption was associated with reduced time to first defecation after colorectal and gynecological surgery, reduced length of hospital stay after gynecological surgery, and reduced postoperative ileus after cesarean section and gynecological surgery.

Although the overall risk of bias is high, based on the included studies, this topic is of potential interest. Strengthening the Discussion by providing clearer descriptions of the hypothesized mechanisms of coffee will improve the manuscript and authors’ conclusions.  A few additional comments are listed below.

Specific Comments

  1. Introduction, line 40: Please provide a reference for the first sentence in this paragraph.
  2. Table 3, first row in Comments column: misspelling of defecation
  3. Discussion, line 222: It is unclear how the statement about chlorogenic acid describes a potential mechanism of coffee on postoperative ileus.  There are other properties of chlorogenic acid that can be described as potential mechanism(s).
  4. Discussion, lines 234-236: If the authors are referring to the amount of coffee in this systematic review and meta-analysis, this should be clarified because the 13 studies tested varying amounts of coffee.
  5. Discussion, lines 258-259: The authors can expand on this limitation because the results are not generalizable to all populations based on the studies that were included.  There is also evidence that the compounds contained in coffee may vary by region, bean type, roast, brew method, etc., which could greatly differ across the world.

Author Response

We greatly appreciate your review of our manuscript and the helpful suggestions. Below are our responses to the reviewers’ comments, with a description of the changes made to the manuscript.

Response to the referees

We greatly appreciate your helpful comments and suggestions.

<Reviewer 2>

General Comments

This systematic review and meta-analysis examined the effect of coffee (caffeinated or decaffeinated) consumption on postoperative ileus and length of hospital stay after various types of abdominal surgeries, including colorectal resection, cesarean section, and gynecological resection. Five of the 13 included clinical trials were conducted in Turkey, with additional trials from Bosnia and Herzegovina, Egypt, Germany, Iran, Israel, Japan, Lithuania, and Switzerland. Among the primary outcomes, coffee consumption was associated with reduced time to first defecation after colorectal and gynecological surgery, reduced length of hospital stay after gynecological surgery, and reduced postoperative ileus after cesarean section and gynecological surgery.

Although the overall risk of bias is high, based on the included studies, this topic is of potential interest. Strengthening the Discussion by providing clearer descriptions of the hypothesized mechanisms of coffee will improve the manuscript and authors’ conclusions.  A few additional comments are listed below.

We thank the reviewer for these insightful comments. We have modified the manuscript according to all comments.

Specific Comments

Introduction, line 40: Please provide a reference for the first sentence in this paragraph.

We appreciate your suggestions. As the reviewer suggested, we have added the reference for the first sentence in this paragraph.

“Coffee is the most widely consumed pharmacological substance worldwide [7].” (on page 1, line 39, in the introduction section)

“7. Abalo, R. Coffee and Caffeine Consumption for Human Health. Nutrients. 2021, 13, 2918. doi: 10.3390/nu13092918.” (on page 18, in the reference section)

Table 3, first row in Comments column: misspelling of defecation

We thank the reviewer for the careful review. As the reviewer suggested, we have corrected typos.

Discussion, line 222: It is unclear how the statement about chlorogenic acid describes a potential mechanism of coffee on postoperative ileus.  There are other properties of chlorogenic acid that can be described as potential mechanism(s).

We thank the reviewer for the careful review. As the reviewer suggested, we have added the following limitations to the revised manuscript.

“Chlorogenic acid has beneficial effects on inflammation and pain [41]. Chlorogenic acid has an anti-inflammatory effect by potently inhibiting the production of tumor necrosis factor-α and interleukin-6 by peripheral blood mononuclear cells [42,43]. In addition, chlorogenic acid inhibits edema formation leading to pain and improve pain following inflammatory responses [42].” (on page 8, lines 221-225, in the discussion section)

Discussion, lines 234-236: If the authors are referring to the amount of coffee in this systematic review and meta-analysis, this should be clarified because the 13 studies tested varying amounts of coffee.

We appreciate your suggestions. As the reviewer suggested, we have added the amount of coffee in Table 1.

Discussion, lines 258-259: The authors can expand on this limitation because the results are not generalizable to all populations based on the studies that were included.  There is also evidence that the compounds contained in coffee may vary by region, bean type, roast, brew method, etc., which could greatly differ across the world.

We thank the reviewer for the careful review. As the reviewer suggested, we have added the following limitations to the revised manuscript.

 “Third, our results may not be generalizable to all populations because the compounds in coffee may vary by region, bean type, roast, and brewing method.” (on page 9, lines 260-262, in the discussion section)

Sincerely yours,

Jun Watanabe, on behalf of the authors.

Department of Surgery, Division of Gastroenterological, General and Transplant Surgery, Jichi Medical University, 3311-1 Yakushiji, Shimotsuke City, Tochigi, 329-0498, Japan

m06105jw@jichi.ac.jp

Tel: +81-285-58-7371, Fax: +81-285-44-3234
